# LbCas12a-D156R Efficiently Edits *LOB1* Effector Binding Elements to Generate Canker-Resistant Citrus Plants

**DOI:** 10.3390/cells11030315

**Published:** 2022-01-18

**Authors:** Hongge Jia, Yuanchun Wang, Hang Su, Xiaoen Huang, Nian Wang

**Affiliations:** Citrus Research and Education Center, Department of Microbiology and Cell Science, Institute of Food and Agricultural Sciences (IFAS), University of Florida, Lake Alfred, FL 33850, USA; honggejia@ufl.edu (H.J.); wangyuanchun@ufl.edu (Y.W.); hang.su@ufl.edu (H.S.); alanwong@ufl.edu (X.H.)

**Keywords:** ttLbCas12a, citrus, *Xanthomonas*, genome editing, citrus cancer, *LOB1*

## Abstract

Citrus canker caused by *Xanthomonas citri* subsp. citri (Xcc) is an economically important disease in most citrus production regions worldwide. Xcc secretes a transcriptional activator like effector (TALE) PthA4 to bind to the effector binding elements (EBEs) in the promoter region of canker susceptibility gene *LOB1* to activate its expression, which in turn causes canker symptoms. Editing the EBE region with Cas9/gRNA has been used to generate canker resistant citrus plants. However, most of the EBE-edited lines generated contain indels of 1–2 bp, which has higher possibility to be overcome by PthA4 adaptation. The adaptation capacity of TALEs inversely correlates with the number of mismatches with the EBE. LbCas12a/crRNA is known to generate longer deletion than Cas9. In this study, we used a temperature-tolerant and more efficient LbCas12a variant (ttLbCas12a), harboring the single substitution D156R, to modify the EBE region of *LOB1*. We first constructed GFP-p1380N-ttLbCas12a:LOBP, which was shown to be functional via Xcc-facilitated agroinfiltration in Pummelo (*Citrus maxima*) leaves. Subsequently, we stably expressed ttLbCas12a:LOBP in Pummelo. Eight transgenic lines were generated, with seven lines showing 100% mutations of the EBE, among which one line is homozygous. The EBE-edited lines had the ttLbCas12a-mediated deletions of up to 10 bp. Importantly, the seven lines were canker resistant and no off-targets were detected. In summary, ttLbCas12a can be used to efficiently generate biallelic/homozygous citrus mutant lines with short deletions, thus providing a useful tool for the functional study and breeding of citrus.

## 1. Introduction

Citrus is one of the top fruit crops worldwide. However, citrus production faces many biotic and abiotic challenges including citrus huanglongbing and bacterial canker, droughts, flooding, and freezes [1,2,3,4,5]. CRISPR-mediated genome editing is promising in dissecting the genetic determinants for improving citrus fruit quality and yield and resistance against biotic and abiotic stresses, and in precision breeding [6,7,8,9]. Jia and Wang [8] first adapted SpCas9/gRNA from *Streptococcus pyogenes* to edit citrus *PHYTOENE DESATURASE* (*PDS*) via transient expression in *Citrus sinensis*. Transgenic expression of SpCas9/gRNA [7,9,10,11,12,13,14,15], SaCas9/gRNA from *Staphylococcus aureus* [16], and LbCas12a/crRNA from *Lachnospiraceae bacterium* [17] have been used for citrus genome editing. Both Huang et al. [7] and Dutt et al. [6] demonstrated the genome editing of citrus protoplasts using SpCas9/gRNA.

CRISPR-mediated genome editing has been used to generate canker resistant citrus varieties. Citrus canker is an economically important citrus bacterial disease that is present in most citrus producing countries. Citrus canker is caused by *Xanthomonas citri* subsp. citri (Xcc). Xcc causes typical hypertrophy and hyperplasia symptoms by secreting a transcriptional activator like effector (TALE) PthA4, a key pathogenicity factor of Xcc, into the nucleus of plant cells. PthA4 binds to the effector binding elements (EBEs) in the promoter region of the canker susceptibility gene *LOB1* to activate its expression, thus causing canker symptoms [18,19,20]. Genome editing of the coding region of *LOB1* abolishes the canker symptoms caused by Xcc [12]. In addition, genome editing of the EBE region also confers citrus resistance to Xcc [10,13,14,21]. Editing the EBE region has advantages over editing the coding region by reducing the putative side effect of mutation of the coding region. In our previous studies, we have generated one homozygous Pummelo (*Citrus maxima*) mutant line containing an adenine deletion within EBE_PthA4_-LOBP, and one biallelic Pummelo mutant line comprising a thymine insertion within one EBE_PthA4_-LOBP allele and a two-adenine deletion within another allele [13]. We have also generated one biallelic ‘Duncan’ grapefruit (*C. paradisi*) mutant line containing a thymine insertion in one allele and an adenine insertion in another allele [21]. These EBE-edited lines are resistant to citrus canker. However, it was reported that TALE effectors are capable of overcoming the disease resistance caused by the mismatches between TALEs and the edited EBE regions [22]. Our data showed that the adaptation capacity of TALEs inversely correlates with the number of mismatches. TALEs harboring seven to nine mismatches were unable to adapt to overcome the incompatible interaction, whereas TALEs that harbored a small number of mismatches (≤5) to the EBE were able to adapt [22]. Thus, it is necessary to generate EBE-edited citrus plants with more mutations. To achieve this goal, we have been using multiple approaches, including increasing the efficacy of the Cas9/sgRNA-based genome editing [10] and using non-Cas9/sgRNA-based tools such as Cas12a, which is known to generate longer deletion than Cas9 [17,23].

Cas12a (formerly Cpf1), derived from *Prevotella* and *Francisella* 1, is a class II/type V CRISPR nuclease. Distinct from Cas9, Cas12a requires T-rich protospacer-adjacent motif (PAM) sequences TTTV (V = A,C,G) and a 23 nt crRNA [24]. Importantly, the distinct PAM of Cas12a from Cas9 enables us to edit different region of the EBE. After cleavage, CRISPR/Cas12a generates 5′ staggered ends distal from PAM [24]. In addition, Cas12a has both DNase activity and RNase activity, which is an advantage for multiplex genome editing [25]. Notably, Cas12a/crRNA was reported to have less off-targets in comparison with CRISPR/SpCas9 [26,27]. Though several Cas12a orthologs have been successfully employed to edit plant genome [28], CRISPR/LbCas12a, derived from *Lachnospiraceae bacterium ND2006*, shows the highest efficacy [29].

LbCas12a has been used to edit genomes of monocot plants, including rice and maize [23,30,31,32,33,34,35,36,37], dicot plants, including Arabidopsis [32,38], tomato [38], lettuce [26], cotton [39], and citrus [17]. However, LbCas12a is less active at lower temperatures [32]. An engineered temperature-insensitive LbCas12a (ttLbCas12a) with a D156R mutation was developed [40]. Intriguingly, LbCas12a-D156R has higher editing activity than LbCas12a in Arabidopsis [41]. At 28 °C, ttLbCas12a markedly outperformed LbCas12a in *Arabidopsis* [41], consistent with the results in tobacco [42].

In this study, we used ttLbCas12a to modify citrus EBE_PthA4_-LOBP. ttLbCas12a function was first tested via Xcc-facilitated agroinfiltration in Pummelo leaf. Subsequently, we conducted stable expression of ttLbCas12a in Pummelo and EBE_PthA4_-LOBP was successfully modified with multiple mutations up to 10 nucleotides deletion. Notably, the ttLbCas12a-mediated mutation rates were 100% in seven transgenic Pummelo lines, including one homozygous line, and the seven edited lines were resistant against citrus canker.

## 2. Materials and Methods

### 2.1. Plasmid Construction

The CmYLCV promoter was amplified using primer CmYLCV-5-*Sbf*I (5′-AGGTCCTGCAGGTGGCAGACATACTGTCCCACAAATGAA-3′), while CmYLCV-3-*Bam*HI (5′-AGGTGGATCCAGCTTAGCTCTTACCTGTTTTCGTCGT-3′) was amplified from Addgene plasmid pDIRECT_10E (Addgene plasmid #91209) and cloned into *Sbf*I-*Bam*HI-digested GFP-p1380N-Cas9 to produce GFP-p1380N-CmYLCV-Cas9. GFP-p1380N-Cas9 was constructed in our previous work [16]. Temperature-tolerant LbCas12a (ttLbCas12a), harboring the single mutation D156R, was obtained from pUC57-ttLbCas12a after *Bam*HI and *Eco*RI digestion (GenScript, Piscataway, NJ, USA). *Bam*HI-LbCpf1-*Eco*RI fragment was inserted into *Bam*HI-*Eco*RI-cut GFP-p1380N-CmYLCV-Cas9 to form GFP-p1380N-ttLbCas12a.

The AtU6-crRNA:LOBP fragment was obtained from pUC57-AtU6-26-crRNA:LOBP (GenScript, Piscataway, NJ, USA). After *Xho*I and *Sac*I cut, AtU6-crRNA:LOBP was inserted into *Xho*I-*Sac*I-digested pUC-NosT-crRNA-cspds from pUC-NosT-crRNA:LOBP. pUC-NosT-crRNA-cspds was constructed previously [17]. Finally, the *Eco*RI-NosT-crRNA:LOBP-NosT-*Pme*I fragment was cloned into *Eco*RI-*Pme*I-cut GFP-p1380N-ttLbCas12a to generate GFP-p1380N-ttLbCas12a:LOBP (Figure 1).

The binary vector GFP-p1380N-ttLbCas12a:LOBP was electroporated into *A. tumefaciens* strain EHA105. Recombinant *Agrobacterium* cells were cultured for Xcc-facilitated agroinfiltration or epicotyl citrus transformation.

### 2.2. Xcc-Facilitated Agroinfiltration in Pummelo

Pummelo (*Citrus* maxima) was grown in a greenhouse at around 28 °C and was pruned for uniform shooting before Xcc-facilitated agroinfiltration. It should be kept in mind that ttLbCas12a performed better at 28 °C [41].

Xcc-facilitated agroinfiltration was performed as described previously with minor modification [43]. Briefly, the fully-expanded young Pummelo leaves were pre-treated with XccΔgumC [44], which was re-suspended in sterile tap water at a concentration of 5 × 10^8^ CFU/mL. Twenty-four hours later, the pre-treated leaf areas were inoculated with *Agrobacterium* cells harboring GFP-p1380N-ttLbCas12a:LOBP or p1380-AtHSP70BP-GUSin. GFP was observed and photographed four days after agroinfiltration. p1380-AtHSP70BP-GUSin was used as a control as described elsewhere [43].

### 2.3. Agrobacterium-Mediated Pummelo Transformation

Pummelo transformation was conducted as described previously with minor modifications [12]. Briefly, Pummelo epicotyl explants were co-incubated with *Agrobacterium* cells harboring the binary vector GFP-p1380N-ttLbCas12a:LOBP. After cocultivation in darkness for 2 or 3 days at 25 °C, the epicotyl explants were placed on regeneration medium at 28 °C, at which ttLbCas12a could edit plant genome more efficiently [41].

All explants were inspected for GFP fluorescence six weeks after incubation. GFP-positive sprouted shoots were selected and micro-grafted on ‘Carrizo’ citrange rootstock plants (*Citrus sinensis* (L.) Osbeck x *Poncirus trifoliata* (L.) Raf.) for further analysis.

The transgenic Pummelo plants were used for PCR analysis with the primers Npt-Seq-5 (5′-TGTGCTCGACGTTGTCACTGAAGC-3′) and 35T-3 (5′-TTCGGGGGATCTGGATTTTAGTAC-3′).

### 2.4. PCR Amplification of Mutagenized LOBP

Genomic DNA was extracted from the Pummelo leaves treated by agroinfiltration or each transgenic Pummelo line. To analyze ttLbCas12a-mediate LOBP mutations, PCR was carried out using primers LOBP3 (5′-AGGTAAGCTTATTCATATTAACGTTATCAATGATT-3′) and LOBP2 (5′-ACCTGGATCCTTTTGAGAGAAGAAAACTGTTGGGT-3′). The PCR products were sequenced either through cloning and colony sequencing or direct sequencing using primer LOB4 (5′-CGTCATTCAATTAAAATTAATGAC-3′). Ten random colonies for each transgenic Pummelo line were selected for sequencing. Chromas Lite program was used to analyze the sequencing results.

### 2.5. GFP Detection

A Zeiss Stemi SV11 dissecting microscope equipped with an Omax camera was used to detect GFP fluorescence of the Pummelo leaves treated by Xcc-facilitated agroinfiltration and GFP-p1380N-ttLbCas12a:LOBP-transformed Pummelo, under illumination of the Stereo Microscope Fluorescence Adapter (NIGHTSEA). Subsequently, the Pummelo leaves were photographed with the Omax Toupview software.

### 2.6. Canker Symptom Assay in Citrus

Wild type and transgenic Pummelo plants were grown in a greenhouse at the Citrus Research and Education Center, University of Florida. Before Xcc inoculation, all plants were trimmed to generate new shoots. Leaves of similar age were inoculated with either Xcc or XccΔpthA4:dLOB1.5 (5 × 10^8^ CFU/mL) using needleless syringes. Canker symptoms were observed and photographed at five DPI.

## 3. Results

### 3.1. Transient Expression of ttLbCas12a to Edit Citrus Genome via Xcc-Facilitated Agroinfiltration

Binary vector GFP-p1380N-ttLbCas12a:LOBP was constructed to edit Pummelo EBE_PthA4_-LOBP (Figure 1). The vector harbors ttLbCas12a, which has the single mutation D156R (Appendix A) [41]. It should be noted that cestrum yellow leaf curling virus (CmYLCV) promoter was used to drive ttLbCas12a expression (Figure 1), since CmYLCV outperformed CaMV 35S and ubiquitin promoter for citrus genome editing [10]. Hammerhead ribozyme (HH) gene was placed at both ends of crRNA to promote editing. In detail, the coding sequence of hammerhead ribozyme and the coding sequence of hepatitis delta virus ribozyme (HDV) were placed at the 5 end and the 3 end of crRNA, respectively (Figure 1b and Appendix A) [33]. In addition, the Pummelo plants were grown at 28 °C, at which ttLbCas12a could edit plant genome with higher efficiency than that at 22 °C [41]. 

As reported previously, Pummelo contains only one kind of LOBP [13], Type II LOBP (Figure 1a and Appendix A). We first tested whether ttLbCas12a functions via Xcc-facilitated agroinfiltration (Figure 2). Among 100 colonies sequenced, one contained ttLbCas12a-directed indels in Pummelo EBE_PthA4_-LOBP (Figure 2), indicating that ttLbC12a is functional for citrus genome editing.

### 3.2. Transgenic Expression of ttLbCas12a in Pummelo

Pummelo epicotyls were transformed with recombinant *Agrobacterium* cells harboring GFP-p1380N-ttLbCas12a:LOBP [45]. Notably, the shoots were generated at 28 °C to facilitate ttLbCas12a-mediated editing. Eight GFP-positive shoots were established (Figure 3), which were designated as #Pum_tt_1 to #Pum_tt_8. The transgenic plants were verified by PCR analysis (Figure 3).

Based on the results of direct sequencing of PCR products, ttLbCas12a-mediated indels took placed in all transgenic Pummelo plants except #Pum_tt_7 (Figure 4). Remarkably, line #Pum_tt_2 is homozygous, since its chromatogram of direct PCR product sequencing has single peaks (Figure 4). Further analysis revealed that ten nucleotides (taaacccctt) were deleted from EBE_PthA4_-LOBP (Figure 5). Furthermore, colony sequencing was employed to calculate the mutation rates. The mutation rates were 100% among the seven transgenic plants (#Pum_tt_ 1–6 and 8), but not in #Pum_tt_7, whose mutation rate was 0 (Figure 5 and Appendix A). The results indicated that ttLbCas12a could modify citrus genome with high efficiency.

### 3.3. Mutation Genotypes of ttLbCpf1 in Transgenic Pummelo

Sanger sequencing results demonstrated that ttLbCas12a deleted ≥2 base pairs (bps) from the target site (Figure 5 and Appendix A). The deletion took place ≥10th bp distal to the PAM site (Figure 5 and Appendix A). Although #Pum_tt_3 had both deletions and insertions (Appendix A), the other six Pummelo lines had only deletion mutation genotypes (Figure 5 and Appendix A), which is consistent with LbCas12a-mediated citrus genome editing [17]. 

### 3.4. Canker Resistance of ttLbCas12a-Transformed Pummelo Plants

Next, we tested whether the eight ttLbCas12a-transformed Pummelo plants were resistant to citrus canker. For this purpose, Xcc was used to inoculate wild type and transgenic Pummelo plants at a concentration of 5 × 10^8^ CFU/mL. No canker symptoms were observed on seven transgenic plants (#Pum_tt_ 1–6 and 8). Typical canker symptoms were observed on wild type and #Pum_tt_7 Pummelo plants at five days post inoculation (DPI) (Figure 6). The results indicated that the homozygous #Pum_tt_2 and the transgenic Pummelo plants (#Pum_tt_1, #Pum_tt_3, #Pum_tt_4, #Pum_tt_5, #Pum_tt_6, and #Pum_tt_8) containing 100% indels were resistant against Xcc infection, which results from EBE_PthA4_-LOBP disruption (Figure 5 and Appendix A).

To further verify that the canker resistance of transgenic Pummelo plants was attributed to EBE_PthA4_-LOBP editing, wild type and transgenic Pummelo plants were treated with XccΔpthA4:dLOB1.5. dLOB1.5 is a designed TALE developed to recognize the sequence 5′ TAAAGCAGCTCCTCCTCATCCCTT 3′ (Appendix A), a sequence in the promoter region of *LOB1* that is different from the EBE_PthA4_ [13]. Sanger sequencing results indicated that there were no modifications in dLOB1.5 binding sites among transgenic Pummelo plants (Appendix A). At 5 DPI, canker symptoms were observed on both wild type and transgenic Pummelo plants inoculated with XccΔpthA4:dLOB1.5 (Figure 6). Taken together, ttLbCas12a-mediated EBE_PthA4_-LOBP modification in the seven transgenic Pummelo plants (#Pum_tt_1, #Pum_tt_2, #Pum_tt_3, #Pum_tt_4, #Pum_tt_5, #Pum_tt_6, and #Pum_tt_8) conferred resistance to citrus canker.

Finally, Cas-Offinder software (Available online: http://www.rgenome.net/cas-offinder/ (accessed on 12 May 2021)) was used to search the potential off-targets of GFP-p1380N-SpCas9p:PumLOBP crRNA. When up to 3 bp mismatches with the targeting crRNA were used for searching, no potential off-targets were identified (Appendix A). Thus, we did not conduct sequencing-based off-target analyses.

## 4. Discussion

In this study, we have successfully generated seven EBE-edited Pummelo plants. ttLbCas12a-mediated mutation genotypes in citrus are distinct from those of SpCas9, which are predominantly short indels (1–2 bp) [9,11,14]. Most of the mutations generated by ttLbCas12a are relatively long deletions, which are similar to those in LbCas12a-transformed citrus and other plants [17,19,23,26,31,32,33,34,35,36,37,38,39]. Different editing features of ttLbCas12a and SpCas9 might result from different cleavage patterns between ttLbCas12a and SpCas9. LbCas12a cleaves DNA at sites distal to the PAM site, leading to 5′ staggered ends, which have 4–5 nucleotide overhangs, whereas SpCas9 cuts DNA 3-4 nucleotides upstream of the PAM site, resulting in blunt ends. In addition, it was reported that the stagger cutting of LbCas12a could lead to the longer deletions [24]. It is worth noting that SpCas9 recognizes NGG PAM, ttLbCas12a recognizes TTTV PAM, and SaCas9 PAM recognizes NNGRRT. All three have been successfully used for genome editing in citrus [8,16,17]. 

Remarkably, the mutation frequencies of seven ttLbCas12a-transformed Pummelo were 100%, and only one plant was not edited. In a previous study, the highest mutation rate was 55% in LbCas12a-transfomred citrus [17]. Thus, ttLbCas12a undoubtedly outperforms LbCas12a for citrus genome editing, which is consistent with the results in Arabidopsis and tobacco [41,42]. However, the mutation efficiency of ttLbCas12-transformed soybean is comparable to that of LbCas12-transformed soybean [46]. This discrepancy might result from different genetic backgrounds of the plants. Although LbCas12a could edit genome in many dicot and monocot plants, the biallelic editing efficacy of LbCas12a remains low. For example, in LbCas12a-transformed rice, the biallelic editing frequencies were less than 50% [29,34,47]. Therefore, it is worth testing whether ttCas12a could improve biallelic editing frequencies in rice and other plants beyond citrus, Arabidopsis, and tobacco. In addition to ttLbCas12a, as one of the most popular genome editing systems, CRISPR/LbCas12a is constantly subjected to improvements, including improved editing efficiency, altered PAM specificities, and multiplexed genome engineering [25,28,41,48,49,50]. 

No off-targets were identified for ttLbCas12a-mediated genome editing of citrus. Intriguingly, similar scenario was also observed for LbCas12a-mediated genome editing of citrus [17]. This concurs with that LbCas12a has lower off-target mutation rates than CRISPR/SpCas9 [50,51]. It should be noted that we only analyzed up to three mismatches. It was reported that increasing mismatches between the on-target and potential off-target sequence significantly decrease the likelihood of off-target effects. The off-target mutation rates decreased from 59% when there is one mismatch between the on-target and off-target sequences to 0% when four or more mismatches are present. Thus, off-target mutations caused by ttLbCas12a are probably low, even though we could not totally rule them out.

In summary,  ttLbCas12a was successfully adapted to generate homozygous/biallelic Pummelo mutant lines; thus, it can be used as a valuable tool for functional study of citrus genes and breeding. Intriguingly, ribonucleoproteins (RNPs) consisting of LbCas12a and crRNA were employed for transgene-free genome editing [26,52]. Delivery of CRISPR/Cas RNPs bypasses the need to remove selection markers from genetically modified plants. It remains to be determined whether ttLbCas12a and LbCas12a RNPs can be used to create transgene-free genome modified citrus.

## Figures and Tables

**Figure 1 cells-11-00315-f001:**
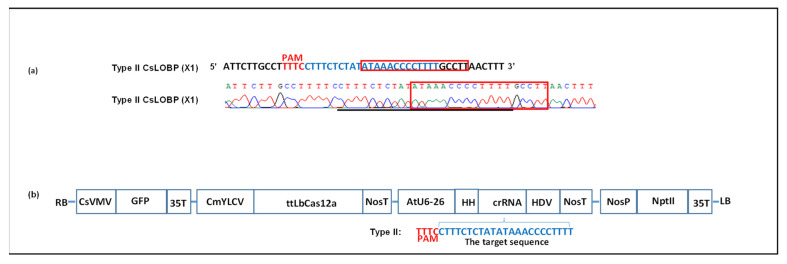
Schematic representation of the binary vector GFP-p1380N-ttLbCas12a:LOBP used to modify Type II LOBP. (**a**) Pummelo Type II LOBP. Part of the Type II LOBP sequence and its chromatogram were presented, in which EBE_PthA4_ was highlighted by red rectangles. A crRNA was designed to target EBE_PthA4_-LOBP, which was indicated by blue. (**b**) Schematic diagram of GFP-p1380N-ttLbCas12a:LOBP. LB and RB, the left and right borders of the T-DNA region; CsVMV, the cassava vein mosaic virus promoter; GFP, green fluorescent protein; 35T, the cauliflower mosaic virus 35S terminator; CmYLCV, the cestrum yellow leaf curling virus promoter; NosP and NosT, the nopaline synthase gene promoter and its terminator; ttLbCas12a, temperature-tolerant LbCas12a containing the single mutation D156R; AtU6-26, *Arabidopsis* U6-26 promoter; target, the 23 nucleotides of Type II LOBP highlighted by blue, was located downstream of protospacer-adjacent motif (PAM); HH, the coding sequence of hammerhead ribozyme; HDV, the coding sequence of hepatitis delta virus ribozyme; NptII, the coding sequence of neomycin phosphotransferase II.

**Figure 2 cells-11-00315-f002:**
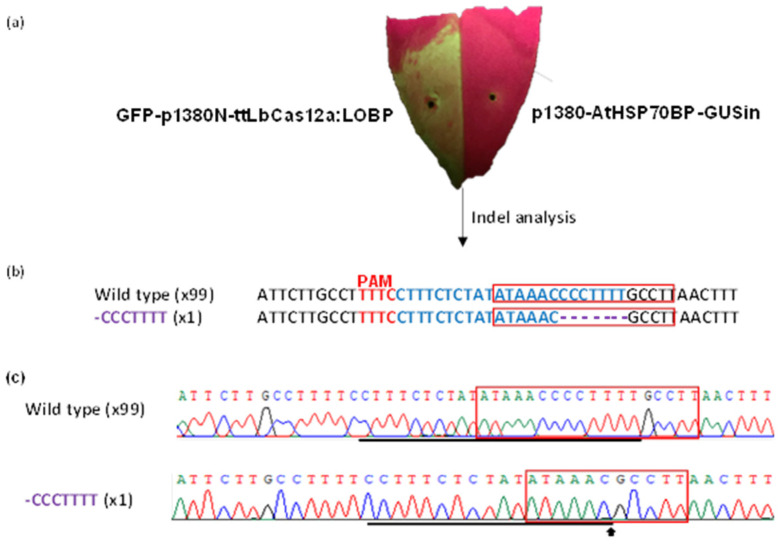
GFP-p1380N-ttLbCas12a:LOBP-directed indels in Pummelo leaf via Xcc-facilitated agroinfiltration. (**a**) Pre-treated with *XccgumC:Tn5*, Pummelo leaf was agroinfiltrated with *Agrobacterium* cells harboring GFP-p1380N-ttLbCas12a:LOBP. After four days, GFP fluorescence was observed. *Agrobacterium* cells harboring p1380-AtHSP70BP-GUSin was used as a negative control. (**b**) ttLbCas12a-directed modification of LOBP. The targeted sequence was shown in blue, and the mutations were shown in purple. (**c**) The representative chromatograms of EBE_PthA4_-TII LOBP and its mutations. The targeted sequence within LOBP was underlined by black lines, and the mutant site was indicated with an arrow. EBE_pthA4_-TII LOBP was highlighted by red rectangles.

**Figure 3 cells-11-00315-f003:**
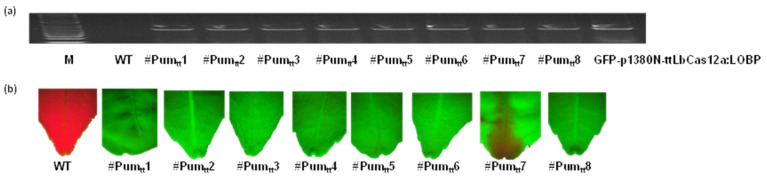
Analysis of GFP-p1380N-ttLbCas12a:LOBP-transformed Pummelo plants. (**a**) PCR was utilized to analyze eight GFP-p1380N-SpCas9p:PumLOBP-transformed Pummelo plants (from #Pum_tt_1 to #Pum_tt_8) with a pair of primers Npt-Seq-5 and 35T-3. The wild type Pummelo (WT) and plasmid GFP-p1380N-ttLbCas12a:LOBP were used as controls. (**b**) GFP fluorescence was observed in transgenic Pummelo plants, whereas wild type plant did not show GFP.

**Figure 4 cells-11-00315-f004:**
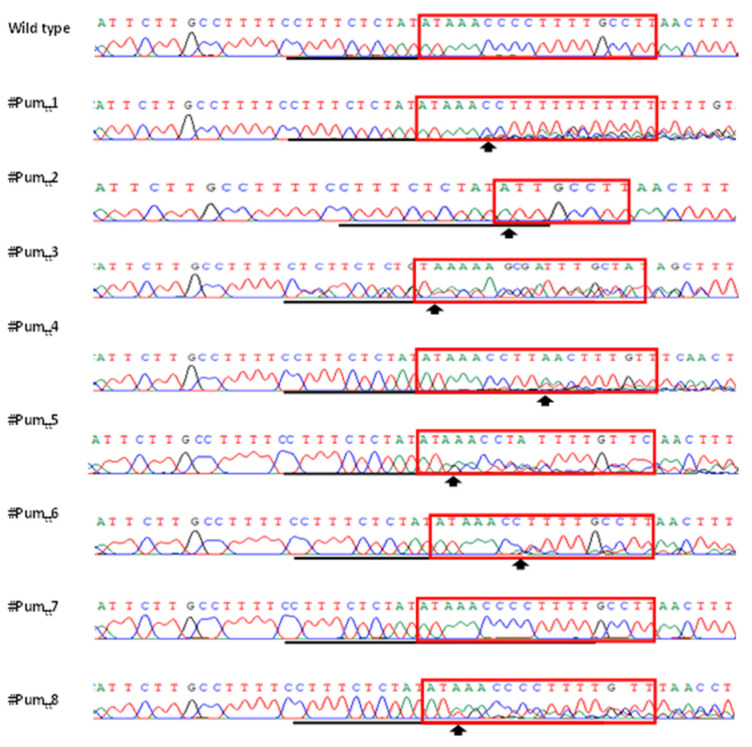
Detection of genome editing of GFP-p1380N-ttLbCas12a:LOBP-transformed Pummelo by direct sequencing of PCR products. The chromatograms of direct sequencing of PCR products. Primers LOBP2 and LOBP5 were used to amplify LOBP from wild type and transgenic Pummelo. Direct sequencing primer was LOB4. The mutation site or the beginning sites of double/multiple peaks were shown by arrows. The targeted sequence was underlined by black lines and EBE_PthA4_-TII LOBP was highlighted by red rectangles.

**Figure 5 cells-11-00315-f005:**
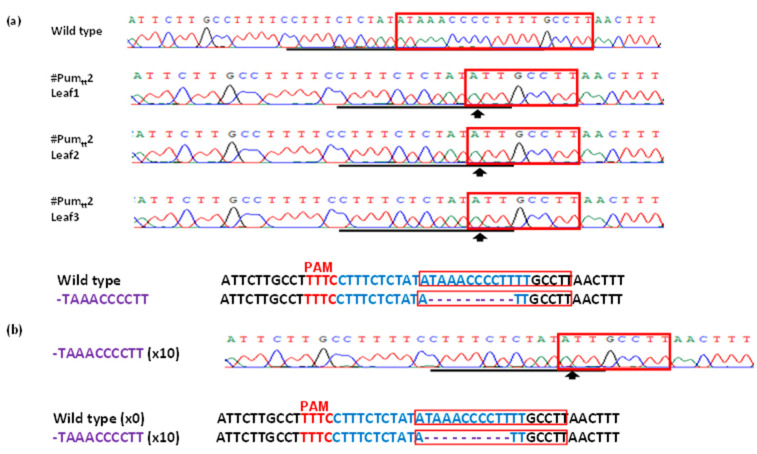
Homozygous line #Pum_tt_2. (**a**) Direct sequencing of PCR product of #Pum_tt_2. Upper: The chromatograms of three different leaves from #Pum_tt_2 were shown. The chromatograms are consistent with one another, which verified #Pum_tt_2 to be homozygous. EBE_PthA4_-LOBP was highlighted by red rectangles. The targeted sequence was underlined by black lines, and the mutation sites were indicated with arrows. The chromatogram of wild type Pummelo plant was included for comparison purpose. Lower: The targeted sequence is shown in blue, and the mutations are shown in purple. (**b**) Sanger sequencing results of #Pum_tt_2. Among 10 colonies sequenced, all of them are taaacccctt deletion. Type II LOBP in #Pum1. A part of LOBP sequences and its chromatogram are shown.

**Figure 6 cells-11-00315-f006:**
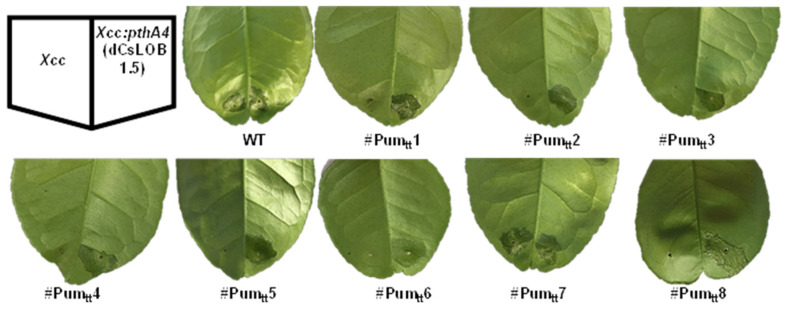
Canker-resistance in ttLbCas12a:LOBP-transformed Pummelo plants. Five days post Xcc inoculation, citrus canker symptoms were observed on wild type Pummelo and #Pum_tt_7, whereas no canker symptoms were observed on other LOBP-edited Pummelo plants, which could be attributed to 100% mutation rates in #Pum_tt_1, #Pum_tt_2, #Pum_tt_3, #Pum_tt_4, #Pum_tt_5, #Pum_tt_6, and #Pum_tt_8. As expected, *XccpthA4:Tn5*(dCsLOB1.5) caused canker symptoms on all plants. dCsLOB1.5 induces *LOB1* to cause canker symptoms by recognizing a different region from EBE_PthA4_-TII LOBP.

## Data Availability

All data are included in this published article (and its Appendix A).

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
