# Peer review of "LbCas12a-D156R Efficiently Edits LOB1 Effector Binding Elements to Generate Canker-Resistant Citrus Plants"

_cells, 2022, doi:10.3390/cells11030315_

Round 1
Reviewer 1 Report
The manuscript revolves around an update of citrus gene editing using CRIPR technologies, this time by implementing ttLbCas12a in Pummelo plants. Edition were performed either by agroinfiltration or stable transgenic line generation. Seven out of 8 lines were efficiently edited in the EBE region of LOB1 promoter.
I only have a pair of concerns to be addressed before publication:
- How do you calculate mutation rate? (I had not access to supplementary figures).
- Pictures of symptoms in Figure 6 are not really clear. In my opinion they require more magnification because only darker patches of Xcc infiltration can be visualized.
- Page 2, lane 83: change maze to maize.
Author Response
The manuscript revolves around an update of citrus gene editing using CRIPR technologies, this time by implementing ttLbCas12a in Pummelo plants. Edition were performed either by agroinfiltration or stable transgenic line generation. Seven out of 8 lines were efficiently edited in the EBE region of LOB1 promoter.
Answer: We appreciate your constructive and valuable comments and suggestions.
I only have a pair of concerns to be addressed before publication:
- How do you calculate mutation rate? (I had not access to supplementary figures).
Answer: We used Sanger sequencing results to calculate mutation rate. In details, PCR was carried out using a pair of primers flanking EBEPthA4 to analyze ttLbCas12a-mediate CsLOBP mutation rate in transgenic pummelo. After purification, the PCR products were subjected to ligation with pMiniT 2.0 (NEB). After transformation, ten random colonies for each transgenic line were chose for Sanger sequencing. As results, all colonies sequenced from #Pumtt1, #Pumtt2, #Pumtt3, #Pumtt4, #Pumtt5, #Pumtt6 and #Pumtt8 were mutant, whereas there was no mutant colony from #Pumtt7. All related data were presented as Supplementary Figure 3, Supplementary Figure 4, Supplementary Figure 5 and Supplementary Figure 6. Sorry for the inconvenience that you had not access to supplementary figures.
- Pictures of symptoms in Figure 6 are not really clear. In my opinion they require more magnification because only darker patches of Xcc infiltration can be visualized.
Answer: We fully agreed that more magnification could show more detailed change in Figure 6. However, the present pictures of symptoms in Figure 6 are clear enough to see canker symptoms. Actually, it is the darker patches that are the canker symptoms, which were not caused by Xcc infiltration. Commonly, The darker patches of Xcc infiltration only last for 2-3 hours in green house. The pictures of symptoms in Figure 6 were taken 5 days post Xcc infiltration.
- Page 2, lane 83: change maze to maize.
Answer: Changed. Thank you.
Reviewer 2 Report
The manuscript by Jia et al. titled “LbCas12a-D156R efficiently edits the promoter region of canker susceptibility gene LOB1 with long deletion to generate canker resistant citrus plants” reported progress of citrus canker resistance through LbCas12a-D156R-targeted modification of the CsLOB1 promoter in citrus. The paper showed efficiency of LbCas12a-D156R in citrus genome editing. Using this system, the authors produced canker resistant plants. The manuscript, however, suffers from a few problems which I would like to point out.
- In the legend of Figure 1 “HH, hammerhead ribozyme; HDV, hepatitis delta virus ribozyme”, hammerhead ribozyme should be reworded as the coding sequence of hammerhead ribozyme. hepatitis delta virus ribozyme should be reworded as the coding sequence of hepatitis delta virus ribozyme.
- Line 190: the sentence “Ribozymes were placed at both ends of crRNA to promote editing” should be reworded as "Ribozyme genes were placed at both ends of crRNA to promote editing". What Ribozymes are used here? Explain it.
- Line 261: I can not find Figure 6a in the manuscript.
- Still now, several CRISPR systems has been successfully used to citrus genome editing. Please briefly compare them in the discussion section.
- Based on the reported citrus genome editing, I think LbCas12a-D156R has no significant advantage in long deletion, thus , I suggest that the title “LbCas12a-D156R efficiently edits the promoter region of canker susceptibility gene LOB1 with long deletion to generate canker resistant citrus plants” is rephrased as “LbCas12a-D156R efficiently edits the promoter region of canker susceptibility gene LOB1 to generate canker resistant citrus plants”.
Author Response
Comments and Suggestions for Authors
The manuscript by Jia et al. titled “LbCas12a-D156R efficiently edits the promoter region of canker susceptibility gene LOB1 with long deletion to generate canker resistant citrus plants” reported progress of citrus canker resistance through LbCas12a-D156R-targeted modification of the CsLOB1 promoter in citrus. The paper showed efficiency of LbCas12a-D156R in citrus genome editing. Using this system, the authors produced canker resistant plants.
Answer: Thank you very much for your constructive and valuable comments and suggestions.
The manuscript, however, suffers from a few problems which I would like to point out.
- In the legend of Figure 1 “HH, hammerhead ribozyme; HDV, hepatitis delta virus ribozyme”, hammerhead ribozyme should be reworded as the coding sequence of hammerhead ribozyme. hepatitis delta virus ribozyme should be reworded as the coding sequence of hepatitis delta virus ribozyme.
Answer: Revised. Thank you.
- Line 190: the sentence “Ribozymes were placed at both ends of crRNA to promote editing” should be reworded as "Ribozyme genes were placed at both ends of crRNA to promote editing". What Ribozymes are used here? Explain it.
Answer: Reworded and explain the ribozymes. Thank you.
- Line 261: I can not find Figure 6a in the manuscript.
Answer: It should be Figure 6 and was corrected. Sorry for the typographical error.
- Still now, several CRISPR systems has been successfully used to citrus genome editing. Please briefly compare them in the discussion section.
Answer: SpCas9, SaCas9 and LbCas12a recognize different PAMs in citrus. We briefly compared them in the discussion section.
- Based on the reported citrus genome editing, I think LbCas12a-D156R has no significant advantage in long deletion, thus , I suggest that the title “LbCas12a-D156R efficiently edits the promoter region of canker susceptibility gene LOB1 with long deletion to generate canker resistant citrus plants” is rephrased as “LbCas12a-D156R efficiently edits the promoter region of canker susceptibility gene LOB1 to generate canker resistant citrus plants”.
Answer: Revised as suggested. Thank you.
Reviewer 3 Report
Is 5 DPI the best choice for citrus canker resistance assay? I suggest evaluating citrus canker resistance at the more representative 10 dpi.
Author Response
Answer: Commonly, there are clear canker symptoms at 4 or 5 DPI in green house. Here we presented the results of 5 DPI. In term of evaluation of symptom development, despite some symptom differences between 5 and 10 DPIs, the results are consistent between the two time points.